# OpenReview forum: "Balanced Learning for Domain Adaptive Semantic Segmentation"
_ICLR.cc/2025/Conference — Submitted to ICLR 2025_

### Official Review · Reviewer_K43a · 2024-10-26

**Soundness:** 3
**Presentation:** 3
**Contribution:** 3
**Rating:** 6
**Confidence:** 5

**Summary:**

This paper introduces a BLDA method to address class-imbalanced problem in unsupervised domain adaptive semantic segmentation. BLDA analyzes the distribution of predicted logits to assess class prediction bias and proposes an online logits adjustment mechanism to balance class learning in both source and target domains. The method incorporates Gaussian Mixture Models (GMMs) to estimate logits distributions and aligns them with anchor distributions using cumulative density functions. Extensive experiments on standard UDA semantic segmentation benchmarks demonstrate significant performance improvements.

**Strengths:**

1. The class-imbalanced is an important issue in DASS, and this paper provides a novel method to tackle this problem by aligning the logits distributions of all classes with anchor distributions to achieve balanced prediction.

2. Extensive experiments have demonstrated the effectiveness of the proposed method.

**Weaknesses:**

1. The paper claims a key contribution in proposing a post-hoc class balancing technique to adjust the network's predictions by establishing two anchor distributions, $P_p$ for positive predictions and $P_n$ for negative predictions. However, the paper lacks sufficient explanation regarding the selection criteria for these anchor distributions, which raises questions about the method's validity and soundness.

2. The current approach in this paper aligns the positive and negative distributions to anchor distributions as part of the post-hoc class balancing strategy. However, based on my understanding, this alignment may not effectively address label noise—a crucial aspect of self-training where pseudo label denoising is often central to performance improvement. Instead, recent studies [1,2] have demonstrated the utility of negative pseudo labeling, showing that leveraging negative information more directly can enhance model robustness and reduce noise. Clarification on the rationale for this alignment-based approach, especially in comparison to existing negative pseudo-labeling methods, would help to justify the method’s efficacy and theoretical basis in the context of label noise mitigation.

[1]. Domain Adaptive Semantic Segmentation without Source Data

[2]. A Curriculum-style Self-training Approach for Source-Free Semantic Segmentation

**Questions:**

## Some questions in Figure 3:

1. Figure 3 presents the logit distributions for positive and negative samples; however, the lack of labeled x- and y-axes in the figure makes it challenging to interpret these distributions effectively.
2. There is no clear explanation of the direction of reweighting and resampling applied to the logit distributions. This omission makes it difficult to understand the intended insights from Figure 3, as well as the overall method’s mechanism and impact on balancing.

3. There are a few grammatical errors, such as the "Discusiion" in L307.

---

### Official Review · Reviewer_Z47K · 2024-11-04

**Soundness:** 2
**Presentation:** 3
**Contribution:** 2
**Rating:** 5
**Confidence:** 4

**Summary:**

This paper discusses the unsupervised domain adaptation problem in semantic segmentation tasks. The method first identifies unbalanced classes by analyzing the predicted logits. Then, it aligns the distributions using a preset anchor distribution. Finally, it also adopts a Gaussian mixture model to estimate logits online to generate unbiased pseudo-labels for self-training. Experiments are conducted on the classic GTAv/SYNTHIA to Cityscapes benchmark for evaluation.

**Strengths:**

1. The paper is well-written and easy to follow. The figures clearly show the distribution trends to help understand the core idea.
2. There are many formula languages to describe the proposed method precisely.
3. The experiments on the GTAv/SYNTHIA/Cityscapes benchmark show clear improvements over baseline methods.

**Weaknesses:**

1. The novelty is limited. The data distribution problem is not newly recognized, and the proposed method adopting anchor distributions for alignment and GMM for unbiased generation is also explored by previous methods. For example, the following papers [a-d] also adopt anchors and/or GMM methods to cross-domain alignment. Please consider providing more discussion with these related works.
2. The method is only verified on a relatively small-scale benchmark. The compared works are from two years ago, which cannot prove this work's value to today's more advanced semantic segmentation approaches. Please consider providing more analysis with other datasets to prove the generalization ability of the method. Optional datasets such as Vistas, IDDA, BDD100k, and VIPER.

[a] Multi-Anchor Active Domain Adaptation for Semantic Segmentation

[b] Category Anchor-Guided Unsupervised Domain Adaptation for Semantic Segmentation

[c] ProtoGMM: Multi-prototype Gaussian-Mixture-based Domain Adaptation Model for Semantic Segmentation

[d] Uncertainty-aware Pseudo Label Refinery for Domain Adaptive Semantic Segmentation

**Questions:**

Please refer to the weaknesses for details. Due to the concerns of the novelty and potential impact, the reviewer is inclined to rate a borderline reject.

---

### Official Review · Reviewer_zNzB · 2024-11-06

**Soundness:** 3
**Presentation:** 3
**Contribution:** 3
**Rating:** 6
**Confidence:** 3

**Summary:**

This paper addresses the challenge of class imbalance in unsupervised domain adaptation (UDA) for semantic segmentation, where labeled source data is used to improve the model’s performance on an unlabeled target dataset. The authors propose a Balanced Learning for Domain Adaptation (BLDA) technique that aligns class predictions by analyzing and adjusting predicted logit distributions, even without prior knowledge of distribution shifts. BLDA enhances UDA model performance by mitigating class bias, particularly for under-represented classes, leading to more accurate segmentation.

**Strengths:**

- The motivation is clear, with a thorough statistical analysis of the class bias issue in unsupervised domain adaptation (UDA) for semantic segmentation (Figures 1 and 2).
- The paper is generally well-written, well-structured, and easy to follow.
- The proposed method comprises four modules. Although each module is simple and widely used in the machine learning field (e.g., GMM and alignment with anchor distributions), these techniques are effective in addressing issues found in this task.
- The experiments are comprehensive, covering three transfer tasks for segmentation, an additional image classification task (included in the supplementary materials), and extensive qualitative analyses.

**Weaknesses:**

1. The proposed method is computationally heavy, as it includes an additional regression head with extra training objectives and requires GMM updates via EM algorithms. Consequently, this approach may incur significantly more computation time and memory usage than baseline methods.

2. In Tables 1, 2, and 4, all existing methods equipped with BLDA are outdated. It remains questionable whether current SOTA methods (in 2023 and 2024) are sufficient to address prediction bias issues.

**Questions:**

1. For weakness 1, could you conduct a theoretical complexity analysis comparing the proposed BLDA with the baseline? Additionally, please report and analyze the actual inference time, training time, and memory usage, along with a comparison to baseline methods (without adding BLDA).

2. For weakness 2, could you integrate BLDA into recent UDA segmentation methods [A], [B], [C], and [D]?

3. The mentioned works are highly relevant but lack citations in this paper. Could you update Section 2.1 (Related Work) to include all necessary references?

[A] Focus on Your Target: A Dual Teacher-Student Framework for Domain-adaptive Semantic Segmentation
[B] CDAC:Cross-domain Attention Consistency in Transformer for Domain Adaptive Semantic Segmentation
[C] Diffusion-based Image Translation with Label Guidance for Domain Adaptive Semantic Segmentation
[D] Learning Pseudo-Relations for Cross-domain Semantic Segmentation

---

> ### Comment · Reviewer_zNzB · 2024-11-26
>
> Thank you for your detailed response. Most of my concerns have been addressed, and I will therefore maintain my current positive rating.

---

### Official Review · Reviewer_9gg7 · 2024-11-07

**Soundness:** 3
**Presentation:** 4
**Contribution:** 3
**Rating:** 5
**Confidence:** 4

**Summary:**

This paper proposes a novel approach called BLDA to address class bias in domain adaptation for semantic segmentation tasks. It first evaluates prediction bias across different classes by analyzing the network's logits distribution. Then, a a post-hoc method is designed to adjust logits distributions after training. With the logits changes, a real-time logits values adjustment module is proposed by using GMMs to estimate logits distribution parameters online. The author then introduces cumulative density estimation as shared structural knowledge to connect the source and target domains. An additional regression head in the network predicts the cumulative distribution value of samples, which represents class discriminative capability, further enhancing adaptation performance on semantic segmentation tasks. The results in the experiments shows its effectiveness as a module addition to selected existing DA for segmentation baselines.

**Strengths:**

1. This paper provide a new way to measure the class distribution changes in semantic segmentation by the logits distribution.
2. The proposed module could easily be applied to existing UDA for semantic segmentation methods, potentially have a broad use in this area.
3. The proposed module is generally effective on most of the classes in the two benckmark tasks.
4. The visual aid is good, provide an intuition of the motivation, also demostrates the effectiveness of the proposed module.

**Weaknesses:**

1. The proposed method relies on the logits distribution. However, this distribution can be affected by data quality and model architecture, which can affect the accuracy of bias assessment.
2. As a DA for segmantation task, a very severe issue is its efficiency concern. Adaptation process already cost a lot of time and computational resources, the proposed method seems exacerbated this issue by multiple GMMs. An efficiency study including wall-clock time or other efficiency measurement will be good to discuess the trade-offs between class-balanced performance and the actual cost.
3. If the anchor distribution is far away from the true distribution of the target domain, logits alignment may be suboptimal, meaning if the domain gap is large, this part may be not work.
4. As a module proposed rather than a whole algorithm, its effectiveness is expected to be confirmed on a considerable large amount of baselines methods, however, only few of them are studied and compared only for Transformer-based methods. I would recommand to evaluate on more baselines such as [1][2][3] and backbones (such as Deeplab v2 Deeplab V3+, for methods such as ProDA) to conform its effectiveness. especially those even have more severe class-imbalance issues.
5. There exist a huge amount of methods or loss functions targeting class-imbalanced issue (for or not for semantic segmentation), some need in related works and some need a experiments for comparison, but only few of them listed and discussed.
6. Since the classes have been categorised as over/under predicted, group them in the experiments and study would be better to understand the module effectiveness on classes with different characteristics.

I will scoring up or down based on the author's reply.

[1]. Domain adaptive semantic segmentation by optimal transport

[2]. DiGA: Distil to Generalize and then Adapt for Domain Adaptive Semantic Segmentation

[3]. Prototypical contrast adaptation for domain adaptive semantic segmentation

**Questions:**

See the weakness section.

---

### Meta-Review · Area_Chair_z4DV · 2024-12-11

**Metareview:**

The work proposes a novel approach namely BLDA to tackle class bias in unsupervised domain adaptation for semantic segmentation. The method analyzes logits distributions to assess class imbalance, employs Gaussian Mixture Models (GMMs) to adjust logits online, and utilizes cumulative density estimation to align source and target domains. Extensive experiments demonstrate its effectiveness as a plug-and-play module, with improvements in segmentation performance across diverse datasets and baselines. Strengths of the paper include its clear motivation and comprehensive experimentation. However, the novelty of the proposed approach is somewhat limited due to similarities with prior works that use GMMs or anchor-based approaches. Reviewers also questioned its computational inefficiency and noted a lack of validation on larger or more diverse benchmarks. The authors have proactively addressed most concerns on experiments, yet the core contributions remain marginal to reach the publication bar of ICLR.

**Additional Comments On Reviewer Discussion:**

During the discussion, reviewers raised concerns about novelty, computational cost, and generalizability. Specific issues included the similarity to prior GMM-based methods, insufficient evaluation on recent baselines, and limited benchmarks. The authors responded with detailed explanations, providing theoretical complexity analysis, efficiency improvements, and validation on additional datasets such as VIPER and BDD. These efforts demonstrated the method’s practical applicability and clarified its unique contributions to addressing class bias in UDA.

However, reviewers like 9gg7 and Z47K remained unconvinced, noting the incremental novelty and suboptimal choice of benchmarks. Despite thorough rebuttals and additional experiments, the reviewers keep their initial ratings due to their doubts about the paper's broader impact and relevance to current SOTA methods. These considerations ultimately lead to the decision to reject, while acknowledging the potential of the work with further development and validation.

---

### Decision · Program_Chairs · 2025-01-22

Reject